# The Sensory-Directed Elucidation of the Key Tastants and Odorants in Sourdough Bread Crumb

**DOI:** 10.3390/foods11152325

**Published:** 2022-08-04

**Authors:** Laura Sophie Amann, Oliver Frank, Corinna Dawid, Thomas Frank Hofmann

**Affiliations:** Food Chemistry and Molecular Sensory Science, TUM School of Life Sciences, Technical University of Munich, Lise-Meitner-Straße 34, 85354 Freising, Germany

**Keywords:** sourdough rye bread, flavor reconstitution, sensometabolome, comparative taste profile analysis, sensomics, unified flavor quantitation, high-throughput targeted LC-MS/MS

## Abstract

Sourdough bread is highly enjoyed for its exceptional flavor. In contrast to bread crust, which has been investigated intensively, the knowledge on bread crumb is rather fragmentary. In this study, the taste-active compounds of sourdough bread crumb were identified and quantified. By means of recombination experiments and omission tests, the authentic flavor signature of sourdough rye bread crumb was decoded and recreated with ten key tastants and eleven key odorants. Based on the final taste and aroma recombinants, a fast and sensitive ultra-high-performance liquid chromatography-tandem mass spectrometry (UHPLC-MS/MS) method using stable isotope dilution analysis (SIDA) was developed and validated. Due to prior derivatization using 3-nitrophenylhydrazine (3-NPH), key tastants and odorants in bread crumb could be quantified simultaneously in a single UHPLC run. The identified key flavor compounds in combination with the developed UHPLC-MS/MS method could offer the scientific basis for a knowledge-based optimization of the taste and odor of sourdough bread.

## 1. Introduction

Sourdough bread is associated with sourness as its key flavor characteristic [1,2]. Whereas the flavor of the bread crust is predominantly formed by thermal reactions during the baking process, the flavor of bread crumb is mainly developed by enzymatic reactions during dough fermentation by lactic acid bacteria and yeasts [2,3]. Furthermore, the flavor of sourdough bread crumb is influenced by factors including flour type [4,5] and addition of starter cultures and yeast [6,7,8,9,10], as well as fermentation time and temperature [1,11,12,13].

The aroma-active compounds of the crumb of a three-stage sourdough rye bread have been recently identified by means of gas chromatography–olfactometry, quantified using stable isotope dilution analysis (SIDA) in combination with high-resolution gas chromatography–mass spectrometry (HRGC–MS), and verified using aroma reconstitution tests [2]. In contrast to the odorants, the compounds responsible for the typical taste of sourdough bread crumb are still unknown. Usually, physicochemical parameters, e.g., pH and total titratable acidity (TTA) as well as the fermentation quotient (FQ, molar ratio of lactic acid to acetic acid) were measured to characterize the nonvolatiles in sourdough bread crumb [9,14,15,16]. Most of the quantitative studies investigating sourdough bread crumb focused on the determination of sodium chloride, acetic acid, lactic acid, and ethanol [9,16,17]. Some extended the spectrum of analyzed organic acids by including, e.g., succinic acid and citric acid [14,18]. All these studies were based solely on quantitative data, and no taste reconstitution experiments have been executed to validate the analytical results.

In order to discover the key compounds responsible for the typical taste and aroma signature in raw and processed foods, the aim of the Sensomics research field is to analytically decode the so-called sensometabolome. This is defined as the complete composition of key sensory-active, low molecular weight compounds that trigger the perception of the characteristic taste and odor of a food [19,20]. The Sensomics approach proved to be a powerful toolbox for the effective elucidation of the key flavor compounds in various foods, e.g., morel mushrooms [21], golden chanterelles [22], cooked crustaceans [23], dried scallops [24], and Gouda cheese [20]. In the present study, the Sensomics approach is adapted and accelerated by transferring the recently published “unified flavor quantitation” [25] to the key flavor compounds of bread crumb.

Commonly, the classical approach for the quantitation of taste and aroma compounds is to separately use LC–MS on the one hand and HRGC–MS on the other. Prior to analysis, HRGC–MS usually requires a separation of nonvolatiles from volatiles, for example by means of steam distillation or solvent-assisted flavor evaporation (SAFE) extraction, because GC is limited to analysis of exclusively volatiles [26]. This time-consuming and labor-intensive separation step is not necessary in the method presented here.

As a scientific background, the key tastants and odorants have to be revealed to investigate the impacts of different ingredients and process parameters on the flavor of sourdough bread crumb. Since it is still not clear which metabolites significantly contribute to the authentic taste of sourdough bread crumb, the purpose of this study was to identify and quantify the key taste compounds in sourdough bread crumb, to reveal their sensory impact on basis of a dose-activity relationship, and to verify their taste contribution using taste recombination and omission experiments. As the aroma compounds in rye bread crumb are already known in the literature, the objective was to simplify the total basic aroma recombinant by means of omission and combination experiments to reveal the key odorants. Furthermore, it was the aim to develop and validate a LC–MS/MS method for quantification of both key tastants and odorants of sourdough bread crumb. In the future, this approach will be helpful for controlling the flavor quality independent of time-consuming sensory analyses during the industrial production of breads.

## 2. Materials and Methods

### 2.1. Chemicals

All compounds were commercially obtained from Merck (Darmstadt, Germany) except for the following chemicals: 2-acetyl-2-thiazoline, 2,3-butanedione, butyric acid, (*E*,*E*)-2,4-decadienal, 2-furfuryl mercaptan, hexanal, 4-hydroxy-2,5-dimethyl-3(2*H*)-furanone (furaneol), methional, (*E*)-2-nonenal, 3-methylbutanal, 3-methylbutanol, 3-methylbutyric acid, 2,3-pentanedione, and vanillin (Givaudan, Vernier, Switzerland), citric acid and phenylacetaldehyde (Alfa Aesar, Ward Hill, MA, USA), ferulic acid, lactic acid, and sodium acetate anhydrous (Fluka GmbH, Neu-Ulm, Germany), 3-hydroxypropionic acid (TCI Chemicals, Zwijndrecht, Belgium), 3-nitrophenylhydrazine hydrochloride (Santa Cruz Biotechnology Inc., Dallas, TX, USA), methanol (HPLC grade; J. T. Baker, Griesheim, Germany), potassium hexacyanoferrate(II) trihydrate and zinc sulfate heptahydrate (Riedel-de Haën, Seelze, Germany), sodium l-lactate (PanReac AppliChem BioChemica, Darmstadt, Deutschland), *N*-(3-(dimethylamino)propyl)-*N*′-ethylcarbodiimide hydrochloride (Carl Roth, Karlsruhe, Germany), and all *α*- and *γ*-glutamyl dipeptides (Bachem, Bubendorf, Switzerland). All stable isotope-labeled compounds were purchased from Cambridge Isotope Laboratories (Tewksbury, MA, USA) with the exception of acetonitrile-*d_3_* (ACN-*d_3_*), deuterium oxide (D_2_O), hexanal-*d_12_*, methanol-*d_4_* (MeOD-*d_4_*), and vanillin-*d_3_* (Merck, Darmstadt, Germany), 2,3-butanedione-*d_6_*, butyric acid-*d_4_*, methional-*d_3_*, 3-methylbutanal-*d_2_*, and 3-methylbutyric acid-*d_2_* (CDN Isotopes Inc., Pointe-Claire, QC, Canada), phenylacetaldehyde-*d_5_* (AromaLAB AG, Planegg, Germany), and 3-(trimethylsilyl)propionic-2,2,3,3-*d_4_* acid sodium salt (TMSP-*d_4_*; Euriso-Top, Saint-Aubin, France). Furthermore, (*E*,*E*)-2,4-decadienal-*d_4_* was synthesized and provided by the Leibniz Institute for Food Systems Biology at the Technical University of Munich. Solvents used for LC-MS/MS analyses were of LC-MS grade (Honeywell, Seelze, Germany). Purified water for all experiments was prepared using a Milli-Q Reference A+ Water Purification System (Merck Millipore, Darmstadt, Germany). For sensory analysis, a commercially available bottled water (Evian, Évian-les-Bains, France) with low mineral content was used.

### 2.2. Bread Samples

A rye bread from Ludwig Stocker Hofpfisterei GmbH (Munich, Germany) was used as a reference sample for the taste recombination of bread crumb. The pure rye bread “Pfister Öko-Wilde Kruste” was produced in a traditional three-stage sourdough bread-making process without the addition of lactic acid bacteria or yeast. Based on the reference bread, the taste of crumbs of two other sourdough breads from this bakery, which were produced in the same way, was recreated: mixed-type bread “Pfister Öko-1331” and wheat bread “Pfister Öko-Weizenlaib” (cf. Table 1). At least 10 mm of the bread crust was removed to yield the bread crumb. The crumb of all breads was freshly analyzed or vacuum packed and stored at −20 °C until further use.

### 2.3. Preparation of the Bread Crumb Extract (BCE)

To obtain a bread crumb extract for qualitative and quantitative analysis as well as the taste recombination experiments, a portion (540 g) of each bread crumb was frozen in liquid nitrogen and ground using a GM300 laboratory mill (4000 rpm, 30 s; Retsch GmbH, Haan, Germany). The ground crumb was extracted with methanol/water (70/30, *v*/*v*, 2.5 L) under stirring (150 rpm, 30 min, RT). After filtration, the bread crumb residue was re-extracted twice with methanol/water (70/30, *v*/*v*, 2.5 L) in the same way as described above. The pooled filtrates were separated from the methanol under reduced pressure, and the residual aqueous solution was lyophilized twice to yield the methanol/water bread crumb extract (BCE; termed RBCE for the rye bread, MBCE for the mixed-type bread, and WBCE for the wheat bread). For the reference bread crumb, the extraction was performed in duplicate. After the evaporation of the methanol, the bread crumb residue was freeze-dried. Half of the residue was extracted three times with water (1.25 L) in the same way as detailed above. The different extracts were lyophilized separately. The final bread crumb residue was freeze-dried twice. All extracts and residues were stored hermetically at −20 °C until further use. For sensory evaluation, the BCE of each crumb was dissolved in water in its “natural” concentration and presented to the trained sensory panel. A portion of BCE (0.1 g) was dissolved in water (50 mL) and membrane filtered for analysis of cations, anions, mono-, and disaccharides by means of high-performance ion chromatography (HPIC). For NMR analysis of BCE, the extract was dissolved in D_2_O, and an aliquot (540 µL) was mixed with an aliquot (60 µL) of the NMR buffer prior to spectroscopy. 

### 2.4. Sample Preparation

A portion (1.0 g) of bread crumb was extracted three times with methanol/water (7 mL, 70/30, *v*/*v*) in a beat beater tube (CK28_15 mL; Bertin Instruments, Montigny-le-Bretonneux, France) using a bead beater homogenizer (Precellys Evolution, Bertin Instruments) as detailed in our previous study [27] with the following modifications. After homogenization (6400 rpm, 3 cycles, 30 s each with 30 s breaks, ≤10 °C), equilibration (500 rpm, 1 h, RT), and centrifugation (4000 rpm, 10 min), an aliquot of the pooled supernatants was membrane filtered (0.45 µm; Minisart RC 15, Sartorius, Göttingen, Germany) for direct analysis using LC–MS/MS. The extraction was carried out in triplicate. A SIDA was performed for the quantification of free amino acids, nucleotides, nucleosides, quaternary ammonium compounds, and opines. For this purpose, the according internal standard mixture was added to the bread crumb before the work-up procedure, and the triplicate of the crumb was extracted once. For the oligosaccharide analysis using HPIC, the triple extraction was performed in the same way as described above with the following modifications. After the extraction of the bread crumb (0.5 g) with methanol/water (3.5 mL, 70/30, *v*/*v*), the pooled supernatants were dried with a rotary vacuum concentrator (30 °C, 11.5 h, 800 Pa) to remove the solvents and re-dissolved in water (1 mL). Subsequently, the extracts were purified by means of solid phase extraction (Strata-X-C, Cation Mixed-Mode Polymeric Sorbent, 33 μm, 75–91 Å, 200 mg/3 mL; Phenomenex, Aschaffenburg, Germany) according to the method of Rebholz et al. [28], again vacuum dried, and re-dissolved in water prior to the HPIC measurement. For the analysis using ^1^H NMR spectroscopy, the extraction was executed in the same way as described above with the following modifications: a portion (0.2 g) of bread crumb was extracted three times with MeOD-*d_4_*/D_2_O (1.4 mL, 70/30, *v*/*v*) instead of methanol/water. The extraction (8000 rpm, 3 cycles, 30 s each with 30 s breaks, ≤10 °C) was conducted using bead beater tubes (CKMix_2 mL; Bertin Instruments) that were filled with ceramic balls (1.4 and 2.8 mm i. d.). The pooled supernatants were membrane filtered, and an aliquot (540 µL) was mixed with NMR buffer (60 µL) before the quantitative NMR measurement.

### 2.5. The Quantitation of Basic Taste Compounds Using HPIC

#### 2.5.1. Cations

The quantitative analysis of cations (Na^+^, NH_4_^+^, K^+^, Mg^2+^, Ca^2+^) was performed as reported earlier [27] with HPIC using a ICS-2000 ion chromatography system (Dionex, Idstein, Germany).

#### 2.5.2. Anions

The quantification of anions (Cl^−^, PO_4_^3^^−^) was carried out based on the work of Toelstede and Hofmann [20] with the following modifications. Aliquots (0.4 µL) of the sample extracts were injected into a ICS-5000 capillary ion chromatography system (Dionex) equipped with a DP-5 dual pump, AS-AP autosampler, DC-5 thermal compartment, and an electrochemical detector operating in conductivity mode. The anions were analyzed using an IonPac AS11-HC capillary column (0.4 mm × 250 mm, 4 µm; Dionex) equipped with an IonPac AG11-HC guard column (0.4 mm × 50 mm; Dionex) and an ACES 300 anion suppressor cell. The chromatographic separation was carried out at 30 °C with a flow rate of 0.015 mL/min using a capillary potassium hydroxide eluent generator cartridge (KOH EGC) for the gradient. Starting with 1 mmol/L KOH for 5 min, the KOH concentration was successively increased to 15 mmol/L within 9 min and further increased to 30 mmol/L within 9 min. Within 8 min, it was raised to 60 mmol/L and maintained at this concentration for an additional 4 min. Finally, the KOH concentration was decreased to 1 mmol/L within 0.1 min and kept at a constant level for 9.9 min. Chromeleon software (version 7.2; Dionex) was used for system control and data processing.

#### 2.5.3. Mono- and Disaccharides

The quantitation of mono- and disaccharides (d-fructose, d-galactose, d-glucose, d-lactose, d-maltose, and d-sucrose) was conducted according to a protocol described earlier [20] with the following modifications. Aliquots (10 µL) of the sample extracts were injected into an ICS-5000 analytical ion chromatography system (Dionex) with the same configuration as described above (Section 2.5.2). However, the electrochemical detector operated in pulsed amperometric mode and was equipped with a silver/silver chloride reference electrode and a gold working electrode operating with a standard quadruple waveform. The carbohydrates were separated on a CarboPac PA-1 analytical column (4 mm × 250 mm, 10 µm; Dionex) connected with a CarboPac PA-1 guard column (4 mm × 50 mm; Dionex). Chromatography was performed at 30 °C at a flow rate of 1 mL/min with isocratic elution of water (80%, eluent A) and an aqueous solution of NaOH (1 mol/L, 20%, eluent B) within 15 min. 

#### 2.5.4. Oligosaccharides

In accordance with Rebholz et al. [28], the quantitative analysis of oligosaccharides (maltotriose, -tetraose, -pentaose, -hexaose, -heptaose, -octaose) was carried out on the ICS-5000 analytical ion chromatography system using a CarboPac PA-100 analytical column (4 mm × 250 mm, 8.5 µm; Dionex) connected with a CarboPac PA-100 guard column (4 mm × 50 mm; Dionex).

### 2.6. Quantification of Glycerol Using an Enzymatic Assay

Glycerol was determined using a commercially available enzymatic test kit (Boehringer Mannheim, R-Biopharm AG, Darmstadt, Germany). 

#### 2.6.1. Sample Preparation for UV-Vis Measurements

The bread crumb was frozen in liquid nitrogen and ground using a GM300 laboratory mill (4000 rpm, 30 s; Retsch GmbH). A portion (3.0 g) of the ground bread crumb was extracted with water (70 mL) under stirring (700 rpm, 90 min, RT). A Carrez clarification was performed with the addition of an aliquot (5 mL) of Carrez-I solution (85 mmol/L potassium hexacyanoferrate(II) trihydrate in water) and an aliquot (5 mL) of Carrez-II solution (250 mmol/L zinc sulfate heptahydrate in water). After mixing, the suspension was adjusted to pH 7.5–8.0 with an aqueous sodium hydroxide solution (0.1 mol/L), filled to 100 mL, mixed, and filtered. The sample work-up was conducted in triplicate. The filtrate was used for the ultraviolet-visible (UV-Vis) measurements.

#### 2.6.2. UV-Vis Spectrophotometry

The absorption measurements were carried out using a GENESYS 10S UV-Vis spectrophotometer (Thermo Fisher Scientific, Waltham, MA, USA). The absorption maximum (340 nm) of reduced nicotinamide-adenine dinucleotide (NADH) was used for the UV-Vis measurements because of the stoichiometric ratio of glycerol and converted NADH. The corresponding enzymatic reactions are given in the Appendix A.

### 2.7. Quantitation of Basic Taste Compounds Using LC-MS/MS

The quantification of free amino acids [29,30], *α*- and *γ*-glutamyl dipeptides [23,31], nucleotides and nucleosides [23], organic acids [22], quaternary ammonium compounds, and opines [23,24] was performed using ultra-high-performance liquid chromatography–tandem mass spectrometry (UHPLC–MS/MS) as reported earlier, with some modifications that are listed in the Appendix A. Data acquisition and instrumental control were carried out using Analyst software (version 1.6.3; Sciex, Darmstadt, Germany). Data evaluation was performed using MultiQuant software (version 3.0.2; Sciex).

### 2.8. Quantitation of Bread Crumb Key Taste and Aroma Compounds after Derivatization

#### 2.8.1. Internal Standard/Analyte Stock Solutions and Calibration

An internal standard mixture (cf. Appendix A) consisting of acetic acid-^13^C_2_ (**1**-^13^C_2_, 57,162 µmol/L), 2,3-butanedione-*d_6_* (**2**-*d_6_*, 132 µmol/L), butyric acid-*d_4_* (**3**-*d_4_*, 908 µmol/L), (*E*,*E*)-2,4-decadienal-*d_4_* (**4**-*d_4_*, 13.0 µmol/L), D-fructose-^13^C_6_ (**5**-^13^C_6_, 29,043 µmol/L), l-glutamic acid-^13^C_5_ (**6**-^13^C_5_, 1746 µmol/L), hexanal-*d_12_* (**7**-*d_12_*, 122 µmol/L), methional-*d_3_* (**9**-*d_3_*, 36.9 µmol/L), 3-methylbutanal-*d_2_* (**10**-*d_2_*, 58.5 µmol/L), 3-methylbutyric acid-*d_2_* (**11**-*d_2_*, 348 µmol/L), phenylacetaldehyde-*d_5_* (**13**-*d_5_*, 65.1 µmol/L), vanillin-*d_3_* (**14**-*d_3_*, 257 µmol/L), and 3-hydroxypropionic acid (**15**, 110,579 µmol/L) was prepared in methanol/water (50/50, *v*/*v*). An analyte stock solution for the quantitation of acetic acid (**1**, 9126 µmol/L), 2,3-butanedione (**2**, 35.5 µmol/L), butyric acid (**3**, 202 µmol/L), (*E*,*E*)-2,4-decadienal (**4**, 3.4 µmol/L), D-fructose (**5**, 4894 µmol/L), l-glutamic acid (**6**, 303 µmol/L), hexanal (**7**, 31.8 µmol/L), lactic acid (**8**, 19,804 µmol/L), methional (**9**, 5.6 µmol/L), 3-methylbutanal (**10**, 15.2 µmol/L), 3-methylbutyric acid (**11**, 84.4 µmol/L), (*E*)-2-nonenal (**12**, 3.7 µmol/L), phenylacetaldehyde (**13**, 17.3 µmol/L), and vanillin (**14**, 63.1 µmol/L) was prepared in acetonitrile/water (50/50, *v*/*v*). The exact concentration of each reference compound was verified by quantitative ^1^H NMR [32]. The stock solution was sequentially diluted 1 + 1 with acetonitrile/water (50/50, *v*/*v*), and an aliquot (10 µL) of the internal standard mixture was added to each dilution (990 µL). An aliquot (40 µL) of each standard solution was used for the derivatization. 

#### 2.8.2. Sample Preparation

Two different sample preparations were performed to quantify the key tastants and odorants in bread crumb. For the quantification of the key tastants, an aliquot (71 µL) of the internal standard mixture was added to a portion (0.1 g) of the bread crumb in a bead beater tube (CK28_15 mL). The suspension was filled to a defined volume (7 mL) with methanol/water (70/30, *v*/*v*), and the extraction was carried out using a bead beater homogenizer (6400 rpm, 3 cycles, 30 s each with 30 s breaks, ≤10 °C). After equilibration (500 rpm, 1 h, RT) and centrifugation (4000 rpm, 10 min), an aliquot of the supernatant was membrane filtered before derivatization. 

For the quantification of the key odorants, a portion (0.5 g) of the bread crumb was weighed in a bead beater tube (CKMix_2 mL). An aliquot (15 µL) of the internal standard mixture was added and was filled to a specified volume (1 mL) with methanol/water (70/30, *v*/*v*). The sample was extracted using a bead beater (8000 rpm, 3 cycles, 30 s each with 30 s breaks, ≤10 °C), equilibrated in an orbital shaker (500 rpm, 2 h, RT), and centrifuged (13,400 rpm, 5 min; Minispin centrifuge, Eppendorf, Hamburg, Germany). The supernatant was membrane filtered before derivatization. 

#### 2.8.3. Derivatization Using 3-Nitrophenylhydrazine

Following modified procedures by Han et al. for carboxylic acids [33], short- and branched-chain fatty acids [34], and sugars [35] and by Hofstetter et al. for flavor-active aldehydes, ketones, and organic acids [25], solutions of 3-nitrophenylhydrazine (3-NPH, 200 mmol/L) in acetonitrile/water (50/50, *v*/*v*) and *N*-(3-(dimethylamino)propyl)-*N*′-ethylcarbodiimide (EDC, 120 mmol/L) in acetonitrile/water (50/50, *v*/*v*) containing pyridine (6%) were prepared for derivatization. Aliquots (40 µL) of the standard solutions or membrane-filtered extracts were mixed with 3-NPH solution (20 µL) and EDC solution (20 µL) and then incubated (30 min, 40 °C, 700 rpm) in a thermo-shaker (Thermo-mixer C, Eppendorf). Before the UHPLC–MS/MS analysis, the solutions were filled with acetonitrile/water (50/50, *v*/*v*) to 1 mL in all cases except for the analysis of key odorants, where it was filled to 160 µL.

#### 2.8.4. Validation Experiments

For the recovery experiments, the analytes were spiked at two levels as triplicates into two matrices: a wheat starch-desalted egg white powder mixture (85/15, *w*/*w*) [27] and the exhaustively extracted bread crumb residue prepared as detailed above (cf. Section 2.3). For the recovery experiments of the key tastants, aliquots (400 µL for the low-spiking experiment; 800 µL for the high-spiking experiment) of the analyte stock solution were spiked to the matrix (0.1 g), the internal standard mixture (71 µL) was added, and the volume was filled to 7 mL with methanol/water (70/30, *v*/*v*). The further sample work-up was carried out as described above (cf. Section 2.8.2). For the recovery experiments of the key odorants, aliquots (60 µL for the low-spiking experiment; 120 µL for the high-spiking experiment) of the analyte stock solution were spiked to the matrix (0.5 g). After the addition of the internal standard mixture (15 µL), the volume was filled (1 mL) with methanol/water (70/30, *v*/*v*), and the samples were prepared as described above (cf. Section 2.8.2).

To determine the limit of detection (LOD) and the limit of quantitation (LOQ), the lowest concentrated solution for calibration was further diluted, and the MultiQuant software (Sciex) was used to measure the signal-to-noise ratio. A signal-to-noise ratio of 3 corresponds to the LOD and a signal-to-noise ratio of 10 to the LOQ. A five-times higher LOD/LOQ was estimated for a determination in food samples.

#### 2.8.5. UHPLC-MS/MS Analysis of 3-NPH Derivatives

The quantification of the 3-NPH derivatives was performed using an ExionLC UHPLC (Sciex) connected to a QTRAP 6500+ MS/MS system (Sciex) operating in negative electrospray ionization (ESI^−^) mode. The ExionLC apparatus consisted of two LC pump systems (ExionLC AD pump), an ExionLC degasser, an ExionLC AD autosampler, an ExionLC AC column oven, and an ExionLC controller. Aliquots (2 µL) of the sample extracts were injected into the UHPLC system equipped with a Kinetex C18 column (2.1 mm × 100 mm, 1.7 μm, 100 Å; Phenomenex) and a corresponding guard column. The chromatographic separation was carried out at 40 °C with a flow rate of 0.4 mL/min. Formic acid (0.1%) in the water served as eluent A and formic acid (0.1%) in acetonitrile as eluent B. The gradient elution started with an isocratic step (80% A) for 1 min and was decreased to 0% A within the next 6.5 min. After a 1 min isocratic elution, eluent A was increased to 80% within 0.5 min and remained at this level for 1 min. The mass spectrometer operated in multiple reaction monitoring (MRM) mode (low mass) using the following instrument settings: ion spray voltage (−4500 V), curtain gas (35 psi), source temperature (450 °C), nebulizer gas (55 psi), heating gas (65 psi), collision activated dissociation (−2 V), and entrance potential (−10 V). Data acquisition and instrumental control were performed with the Analyst software (Sciex). The acquired mass spectrometric data were analyzed using the MultiQuant software (Sciex).

### 2.9. Nuclear Magnetic Resonance (NMR) Spectroscopy

All NMR measurements were recorded on a AVANCE III 400 MHz system (Bruker, Rheinstetten, Germany) equipped with an inverse BBI probe. All samples were analyzed in NMR tubes (5 mm × 178 mm; Bruker, Faellanden, Switzerland). For the verification of the exact concentration of each analyte solution, the compounds were dissolved in aliquots (1.5 mL each) of D_2_O (analytes **1**–**3**, **5**, **6**, and **8**), ACN-*d_3_* (**12**), or MeOD-*d_4_* (**4**, **7**, **9**–**11**, **13**, **14**), and aliquots (600 µL each) of these solutions were used for spectroscopy. For the NMR measurements of bread crumb and BCE, a NMR buffer was used. The buffer solution was prepared by dissolving KH_2_PO_4_ (10.2 g), KOH (1.5 g), TMSP-*d_4_* (50 mg), and NaN_3_ (5 mg) in D_2_O (40 mL), adjusting to pH 7.0 with KOH solution (4.0 mol/L in D_2_O), and filling up to 50 mL with D_2_O. The spectra were referenced to the TMSP signal when the NMR buffer was added to the solvent signal (ACN-*d_3_*, D_2_O, MeOD-*d_4_*) and are given in parts per million (ppm). For data processing, TopSpin 3.6 software (Bruker) was used. Quantitative ^1^H NMR spectroscopy was performed by calibrating the spectrometer using external calibration standards of caffeine and tyrosine in D_2_O, by applying the ERETIC II tool (Electronic REference To access In vivo Concentrations) using the PULCON (PULse length based CONcentration determination) methodology and by comparison with the in-house database containing 117 reference taste compounds [32,36]. For the NMR analysis of bread crumb and BCE, quantification was performed under the same conditions as above but with the additional use of the Bruker standard water suppression pulse sequence (1D noesygppr1d) [37].

### 2.10. Dose-over-Threshold Factors

After quantitative analysis, the dose-over-threshold (DoT) factors were calculated as the ratio of concentration and taste recognition threshold to estimate the taste activity of each individual taste compound [21].

### 2.11. Sensory Analysis

#### 2.11.1. General Conditions and Panel Training

The sensory panel was composed of 17 experienced human assessors (aged 23–32 years) who had no history of known taste or aroma disorders and had given informed consent to participate in the following sensory evaluations. All sessions were performed in single booths of an air-conditioned sensory room at 20–22 °C, and the light was adjusted to orange to mask visual differences between the samples. Before the first and between each sensory sample, the panelists rinsed their mouth (≥10 s) with water in the taste tests or neutralized their odor perception by smelling (≥5 s) roasted coffee beans in the aroma experiments. For the taste evaluation, the panelists used nose clips to prevent cross-modal interactions with the aroma compounds. The pH of all solutions was adjusted to that of the corresponding sourdough bread crumb or BCE by adding trace amounts of formic acid. To become familiar with the taste language and impressions, the panelists were trained to identify and assess aqueous reference solutions with salty, sour, sweet, bitter, astringent, and umami taste as reported earlier [27]. Kokumi activity (mouthfulness enhancement) was trained by comparison of the taste perception of a blank model broth (control) with a model broth spiked with reduced glutathione [38,39]. For orthonasal aroma evaluation, a panel training was carried out with aqueous supra-threshold solutions (5 mL) of reference aroma compounds that were presented in covered glass vessels (40 mm i. d.). The subjects practiced recognizing the different odor qualities of the following odorants: acetic acid (1100 mg/L) for sour, 3-methylbutanal (20 µg/L) for malty, 2,3-butanedione (750 µg/L) for buttery, 3-hydroxy-4,5-dimethyl-2(5)*H*-furanone (sotolon; 15 µg/L) for spicy, butyric acid (50 mg/L) for sweaty, (*E*,*E*)-2,4-decadienal (10 µg/L) for fatty, furaneol (3 mg/L) for caramel-like, 2-acetyl-2-thiazoline (90 µg/L) for roasty, and 2-furfuryl mercaptan (20 µg/L) for coffee-like aroma.

#### 2.11.2. Preparation of Taste Recombination Models

A total basic taste recombinant (tRec_rye_) was prepared by dissolving the quantified basic tastants of the rye bread crumb with DoT factors ≥ 0.6 in their “natural” concentrations in water. The pH was adjusted to 4.4 by adding trace amounts of formic acid, which represents the pH of RBCE in water. In addition, an extended total taste recombinant (tRec^+^_rye_) was prepared in water by spiking the amounts of acetic and lactic acid lost during solvent removal by rotary evaporation and lyophilization during the preparation of RBCE (cf. Section 2.3). The pH was adjusted to 4.0, representing the pH of RBC. By exclusively adding the determined key taste compounds of rye bread in their “natural” concentrations, a partial basic taste recombinant (pRec_rye_) was prepared in water (pH 4.4). Furthermore, an extended partial basic recombinant with spiking of acetic and lactic acid (pRec^+^_rye_) was prepared in water (pH 4.0). Both the expanded total and partial recombinant were prepared in matrix (tRec^+^_rye_(matrix), pRec^+^_rye_(matrix)). Thereby, the exhaustively extracted bread crumb residue prepared as described above (cf. Section 2.3.) served as the matrix.

The extended total and partial recombinants of the mixed-type bread and the wheat bread were prepared with the same tastants as detailed above in their “natural” concentrations (cf. Table 2) in water (tRec^+^_mixed_, pRec^+^_mixed_, tRec^+^_wheat_, pRec^+^_wheat_) and in the corresponding exhaustively extracted bread crumb (tRec^+^_mixed_(matrix), pRec^+^_mixed_(matrix), tRec^+^_wheat_(matrix), pRec^+^_wheat_(matrix)), respectively.

#### 2.11.3. Preparation of Aroma Recombination Models

The total basic aroma recombinant was taken from the literature [2]. A 400-times concentrated recombinant solution was prepared in ethanol with the basic odorants of the rye bread crumb (cf. Appendix A). On the one hand, this ethanolic solution was added to citrate buffer (0.01 mol/L), which was adjusted to the pH of the rye bread crumb (pH 4.25) with citric acid, and on the other hand, to wheat starch (Kröner-Stärke GmbH, Ibbenbüren, Germany). The citrate buffer model was stirred (500 rpm, 30 min), and the starch model was homogenized using a shaking machine (30 min; Turbula, Willi A. Bachofen, Basel, Switzerland). The partial basic aroma recombinant was prepared in the same way as mentioned above in citrate buffer and starch. The odor activity values (*OAV*) were calculated on the basis of the odor thresholds either in water (*OAV*_water_) or in starch (*OAV*_starch_), respectively, because both are the main ingredients in bread.

#### 2.11.4. Taste Profile Analysis (TPA)

The panelists were instructed to rate the perceived intensities of the taste descriptors salty, sour, sweet, bitter, astringent, umami, and kokumi on a linear scale from 0 (not perceivable) to 5 (strongly perceivable). For comparative TPA in aqueous solution, the lyophilized rye bread crumb extract dissolved in water in its “natural” concentration without (RBCE) or with spiking of acetic and lactic acid (RBCE^+^) was chosen as a reference. For comparative TPA in matrix, the rye bread crumb (RBC) mixed 1:1 with water served as a reference. All recombination models (tRec, pRec, tRec(matrix), tRec^+^, pRec^+^, pRec^+^(matrix)) were presented to the panelists to validate the fit of the models in comparison to the reference.

#### 2.11.5. Triangle Tests

The omission and combination tests were performed using triangle tests, also known as three-alternative forced choice (3-AFC) tests. Three covered flasks encrypted with a three-digit random number were presented to the panelists in randomized order. Two of the presented samples were the same and one was different. According to the forced-choice method, the panelists were instructed to state the sample that they perceived to be different from the other two samples. The triangle tests were statistically evaluated with a significance table according to ISO 4120 [40]. The minimum number of correct answers that was required to achieve a significant difference on a significance level *α* with a specified number of panelists is outlined in the table. All *p* values ≤ 0.05 were judged as significant.

Omission experiments were conducted to analyze the relevance of the individual compounds to the overall taste or aroma of the rye bread crumb. By means of triangle tests and comparative TPA, incomplete taste models, each lacking an individual taste compound from the tRec, were evaluated using tRec as a reference. Incomplete aroma models were prepared by omitting a single aroma compound or combining two aroma compounds with the same odor quality and a similar chemical structure. For the combination test, the odorant with the lower *OAV*_water_ was omitted, and the concentration of the other odorant was increased according to the sum of the *OAV*_water_ of both odorants. Each of these incomplete aroma models was compared with the total aroma recombinant by means of triangle tests in both model matrices: aqueous citrate buffer solution and starch. 

### 2.12. Statistical Analysis

Statistical significance is depicted as follows: NS = not significant (*p* > 0.05), * significant (0.05 ≥ *p* > 0.01), ** highly significant (0.01 ≥ *p* > 0.001), and *** very highly significant (*p* ≤ 0.001). All data and statistical analyses were performed using OriginPro 2020 (OriginLab Corporation, Northampton, MA, USA) and Excel 2016 (Microsoft Corporation, Redmond, WA, USA). 

## 3. Results and Discussion

### 3.1. Sensory-Directed Decoding and Reconstitution of Bread Crumb Taste

In order to identify and quantify the nonvolatile compounds inducing the characteristic taste of sourdough bread crumb, the so-called Sensomics approach was applied [19,20,21,22,23,24]. At first, the crumb of a rye bread, a mixed-type bread, and a wheat bread were presented to a trained sensory panel who evaluated the individual intrinsic taste profiles (Figure 1). Independent of the kind of sourdough bread, saltiness and sourness were the predominant taste impressions with scores of 2.4, 2.5, and 2.6 (salty) as well as 2.5, 2.1, and 1.5 (sour) for the rye, mixed-type, and wheat bread crumb, respectively. The remaining taste attributes sweet, bitter, astringent, umami, and kokumi were judged with intensities of less than 1 and therefore should have a minor impact on the overall taste of bread crumb. The rye bread crumb exhibited a more pronounced taste compared with the other two crumbs based on the perceived intensities of all taste qualities except for saltiness (cf. Figure 1). Based on these findings, the rye bread crumb served as the reference bread crumb and was chosen for the initial taste recombination experiments. 

To obtain extracts representing the taste profile of bread crumb, the fresh crumb of each bread was milled and extracted three times using methanol/water (70/30, *v*/*v*). The bread crumb residues were further extracted three times with water to produce the exhaustively extracted residues, which were perceived as tasteless. After removing the solvents, the extracts were dissolved in water in their “natural” concentrations as present in the bread crumbs prior to TPA. A comparison of the taste profiles of fresh crumbs and their corresponding extracts revealed similar profiles (cf. Appendix A) with the exception of higher perceived intensities for saltiness (RBCE: 3.0, MBCE: 3.4, WBCE: 3.3) and lower ones for sourness (RBCE: 2.1, MBCE: 1.6, WBCE: 1.3). In a previous study [41], an aqueous NaCl solution was perceived as significantly saltier (*p* = 0.001) than a bread crumb containing the same absolute amount of NaCl. The authors attributed the difference to a slower increase in sodium in the saliva and therefore a less perceivable saltiness contrast before and during the tasting of the bread crumb compared with the NaCl solution [41]. The lower scores for the perceived sourness of the extracts are due to losses of acetic and lactic acid during the solvent removal by means of rotary evaporation and lyophilization. Consequently, the lost amounts of the acids were spiked in subsequent experiments (BCE^+^).

After extraction, the basic tastants in rye bread crumb were identified and quantified with quantitative ^1^H NMR and HPIC as well as UHPLC-MS/MS using SIDA, and the DoT factors were calculated to evaluate the contribution of the individual taste compounds to the bread crumb taste. In total, 23 basic tastants were found with DoT factors equal to 0.1 or higher: sodium, potassium, ammonium, chloride, phosphate, magnesium, calcium, d-fructose, succinic acid, l-glutamic acid, l-aspartic acid, acetic acid, lactic acid, and ferulic acid (DoT ≥ 0.6, cf. Table 2) as well as ethanol, betaine, glycerol, glucose, sucrose, maltose, maltotriose, maltotetraose, and l-alanine (0.6 > DoT ≥ 0.1, cf. Appendix A).

Chemosensory recombination experiments were carried out in order to verify the quantitative data. For this purpose, all taste-active compounds with DoT factors of at least 0.6 were dissolved in water in their “natural” concentrations as present in rye bread crumb to yield the total basic taste recombinant tRec_rye_ (Table 2). The taste of tRec_rye_ was compared with that of RBCE dissolved in water in a comparative TPA and triangle test. The taste profiles of tRec_rye_ and RBCE (cf. Appendix A) coincided well, and the triangle test result showed no significant difference (*p* = 0.17), indicating that tRec_rye_ imitates the taste of the RBCE in a very good way.

To elucidate the key tastants in rye bread crumb, omission experiments were carried out in aqueous solution by omitting a single taste compound of tRec_rye_ (Table 3). No omission tests were conducted with sodium and chloride because they exhibited the highest DoT factors (25.2; 25.8). The omission experiments revealed that ten of the tastants have a significant contribution to the overall taste (*p* ≤ 0.05); thus, they represent the key taste compounds in rye bread crumb: sodium, potassium, ammonium, chloride, magnesium, calcium, d-fructose, l-glutamic acid, acetic acid, and lactic acid. The comparison of the final partial recombinant pRec_rye_, solely containing all key tastants, with the initial total recombinant tRec_rye_ demonstrated that there is no significant difference (*p* = 0.60) between the taste models. This sensory evaluation was repeated for the extended taste models in aqueous solution as well as in matrix (cf. Table 3). The triangle test between RBCE^+^ and tRec^+^_rye_ (*p* = 0.13) as well as between tRec^+^_rye_ and pRec^+^_rye_ (*p* = 0.60) showed no significant differences. For the sensory analysis in matrix, the rye bread crumb (RBC) was mixed with water, and both the extended total and partial taste recombinants were added to the exhaustively extracted residue (tRec^+^_rye_(matrix); pRec^+^_rye_(matrix)). No significant differences were perceived between RBC and tRec^+^_rye_(matrix) (*p* = 0.45) or between tRec^+^_rye_(matrix) and pRec^+^_rye_(matrix) (*p* = 0.66). Consequently, the developed full and partial recombinants represent reliable models for the taste of the sourdough rye bread crumb and can be applied in future studies.

The taste of the mixed-type and wheat bread crumb was reconstituted based on the generated taste models for the rye bread crumb, which served as the reference bread crumb. The total taste recombinants were prepared by dissolving the 14 basic tastants listed in Table 2 in water to match their “natural” concentrations as present in the mixed-type (tRec^+^_mixed_, tRec^+^_mixed_(matrix)) and wheat bread crumb (tRec^+^_wheat_, tRec^+^_wheat_(matrix)). The partial taste recombinants were prepared exclusively using the ten key tastants (pRec^+^_mixed_, pRec^+^_mixed_(matrix), pRec^+^_wheat_, pRec^+^_wheat_(matrix)). Analogous to what is detailed above, the different models were compared by means of triangle tests and comparative TPA (cf. Appendix A). All performed triangle tests results showed no significant differences (*p* > 0.05): MBCE^+^ vs. tRec^+^_mixed_ (*p* = 0.13), tRec^+^_mixed_ vs. pRec^+^_mixed_ (*p* = 0.45), MBC vs. tRec^+^_mixed_(matrix) (*p* = 0.26), tRec^+^_mixed_(matrix) vs. pRec^+^_mixed_(matrix) (*p* = 0.26), WBCE^+^ vs. tRec^+^_wheat_ (*p* = 0.66), tRec^+^_wheat_ vs. pRec^+^_wheat_ (*p* = 0.26), WBC vs. tRec^+^_wheat_(matrix) (*p* = 0.26), and tRec^+^_wheat_(matrix) vs. pRec^+^_wheat_(matrix) (*p* = 0.26). Therefore, the transfer of the taste models from the reference bread crumb to the other bread crumbs was successful: The taste re-engineering experiments demonstrated that the basic tastants, summarized in Table 2, are not only the key compounds imparting the typical taste profile of sourdough rye bread crumb but also of sourdough mixed-type and wheat bread crumb.

Sodium chloride was found to be the quantitatively predominant tastant in sourdough bread crumb (Table 2). The determined NaCl contents are in the same range as reported for sourdough wheat bread crumb [16] as well as for wheat bread, mixed-type bread, and rye bread [43]. The major salt amount found in bread crumb derives from the direct addition of NaCl in the dough-making process. The remaining minerals originate from the other ingredients, flour and tap water. The determined contents of potassium are well in line with levels of 26.5 mmol/kg [44], 33.7 mmol/kg [45], and 31.2–43.8 mmol/kg [43] in wheat bread as well as 34.7 mmol/kg in mixed-type bread and 44.2 mmol/kg in rye bread [43] as reported in former studies. The remaining minerals—ammonium, phosphate, magnesium, and calcium—were present in concentrations lower than 17.0 mmol/kg with maximum DoT factors of 4.3 and therefore had a minor effect on the overall taste of sourdough bread crumb. In addition to NaCl, acetic acid and lactic acid are major tastants in the investigated sourdough bread crumbs, and the determined concentrations are in the same range as previously reported data for ciabatta bread crumb [17] and for sourdough wheat bread [9,14]. In comparison to this, relatively low contents were found in sourdough bread crumb of the umami-tasting organic acids—succinic acid, l-glutamic acid, and l-aspartic acid (cf. Table 2). Ferulic acid, which derives from grain flour, was determined in the same levels as recently published for the crumb of sourdough wheat bread and sourdough rye bread [46]. The determined concentrations of ethanol are well in alignment with previous studies that quantified it in the crumb of baguettes [16] as well as in the crumb of ciabatta breads [17]. In previous investigations of sourdough, many tastants determined here have been reported to play an important role in the sourdough fermentation by lactic acid bacteria and yeasts. While fructose, glucose, maltose, and sucrose serve as carbon sources and are utilized, acetic acid, lactic acid, ethanol, glycerol, and succinic acid are produced during sourdough fermentation [47,48,49].

### 3.2. Sensory-Directed Decoding of Key Odorants in Bread Crumb

In contrast to the nonvolatile taste compounds, the volatile aroma compounds in sourdough rye bread crumb are already known. Kirchhoff and Schieberle [2] performed an activity-guided screening for odorants in the bread crumb using aroma extract dilution analysis (AEDA), followed by structure elucidation, quantitation, and the calculation of the odor activity values as the ratio of the concentration and the odor threshold in both water (*OAV*_water_) and starch (*OAV*_starch_). The aroma of the sourdough rye bread crumb was reconstituted using 20 odorants: acetic acid (**1**), 2,3-butanedione (**2**), butyric acid (**3**), (*E*,*E*)-2,4-decadienal (**4**), hexanal (**7**), methional (**9**), 3-methylbutanal (**10**), 3-methylbutyric acid (**11**), (*E*)-2-nonenal (**12**), phenylacetaldehyde (**13**), vanillin (**14**), 3-methylbutanol (**19**), phenylacetic acid (**20**), 2-phenylethanol (**21**), 2,3-pentanedione (**22**), sotolon (**23**), (*Z*)-4-heptenal (**24**), (*Z*)-2-nonenal, (*E*)-*β*-damascenone, and 4,5-epoxy-(*E*)-2-decenal. The authors carried out omission tests for compounds with low *OAVs* like (*Z*)-2-nonenal, (*E*)-*β*-damascenone, and 4,5-epoxy-(*E*)-2-decenal and found no significant differences (*p* > 0.05) compared with the recombinant containing these odorants [2]. Therefore, these compounds could be omitted, resulting in a basic aroma recombinant containing 17 odorants (cf. Figure 2 compounds **1**–**4**, **7**, **9**–**14**, **19**–**24**; Appendix A).

To elucidate the key odorants in rye bread crumb, further omission experiments and additional combination experiments were performed in the present study (cf. Appendix A). The omission and combination tests were carried out with triangle tests in aqueous citrate buffer solution and starch. The omission tests of 2-phenylethanol (**21**; *OAV*_water_ = 0.7, *OAV*_starch_ = 6), sotolon (**23**; 21, 3), and (*Z*)-4-heptenal (**24**; 14, 0.9) in both model matrices showed no significant differences (*p* > 0.05). As a consequence, these odorants could be omitted. Aroma compounds with a similar chemical structure and the same aroma quality were combined by omitting the aroma compounds with the lower *OAV*_water_ and by enhancing the concentration of the other compound according to the sum of the *OAV*_water_ of both aroma compounds. No significant differences (*p* > 0.05) occurred from combining 2,3-butanedione with 2,3-pentanedione (**2**, **22**; buttery), 3-methylbutanal with -butanol (**10**, **19**; malty), and phenylacetaldehyde with 2-phenylacetic acid (**13**, **20**; honey-like). Hence, only one of the odorants was present in the final partial recombinant (2,3-butanedione, 3-methylbutanal, phenylacetaldehyde), and the other one could be left out (2,3-pentanedione, 3-methylbutanol, 2-phenylacetic acid; cf. Table 4), respectively. Further combination experiments with butyric and 3-methylbutyric acid (**3**, **11**; sweaty) as well as (*E*,*E*)-2,4-decadienal and (*E*)-2-nonenal (**4**, **12**; fatty) revealed a significant difference (*p* ≤ 0.05) in at least one matrix (cf. Appendix A). Based on the results, these aroma compounds could not be combined or substituted. In terminal triangle tests, the final partial aroma recombinant, containing the odorants listed in Table 4, was compared with the initial total aroma recombinant in both matrices. No significant differences (*p* > 0.05) were perceived by the sensory panel, demonstrating that these eleven aroma compounds are the key odorants in rye bread crumb. Consequently, acetic acid essentially affects both the taste and the aroma of sourdough bread crumb.

### 3.3. Method Development and Validation

Based on the identified key flavor compounds in bread crumb, an UHPLC–MS/MS method using SIDA was developed including all key tastants and odorants of bread crumb (cf. Figure 2, compounds **1**–**14**) with ethe xception of the minerals, which were quantified with HPIC. To facilitate sensitive MS/MS detection of taste- and aroma-active carbonyl compounds, solutions of associated reference compounds and bread crumb extracts were derivatized using 3-NPH, and the ion source and ion path parameters in ESI^–^ mode were optimized via syringe-infusion by means of software-assisted ramping (Appendix A). Non-optimized declustering potential values for MRM transitions of analytes **1** and **8** and of their corresponding internal standards (**1**-^13^C_2_ and **15**) were used to prevent saturation due to high concentrations of these analytes in bread crumb. The derivatization of analytes **1**–**5** and **7**–**14** is shown in Figure 3. The figure shows that 3-NPH reacts with both carbonyl groups of 2,3-butanedione (**2**) forming **2**-(3-NPH)_2_. The reactions occurring during the derivatization of glutamic acid (**6**) with 3-NPH are depicted in the Appendix A. In the predominant reaction, glutamic acid cyclizes to pyroglutamic acid, which reacts with 3-NPH, leading to a *m*/*z* value of 264. Because of this, only the sum of glutamic acid and pyroglutamic acid could be determined with the present method.

The presence of acetic acid, 2,3-butanedione, butyric acid, (*E*,*E*)-2,4-decadienal, d-fructose, l-glutamic acid, hexanal, lactic acid, methional, 3-methylbutanal, 3-methylbutyric acid, (*E*)-2-nonenal, phenylacetaldehyde, and vanillin was confirmed in bread crumb by comparison of LC retention times and MS/MS spectra. The separation of the analytes was conducted in a single run within 10 min. Quantitation was performed with SIDA using acetic acid-^13^C_2_, 2,3-butanedione-*d_6_*, butyric acid-*d_4_*, (*E*,*E*)-2,4-decadienal-*d_4_*, d-fructose-^13^C_6_, l-glutamic acid-^13^C_5_, hexanal-*d_12_*, 3-hydroxypropionic acid, methional-*d_3_*, 3-methylbutanal-*d_2_*, 3-methylbutyric acid-*d_2_*, phenylacetaldehyde-*d_5_*, and vanillin-*d_3_* as corresponding internal standards, which were spiked to the bread crumb prior to sample work-up. Instead of stable isotope-labeled lactic acid, 3-hydroxypropionic acid served as the internal standard for lactic acid because it does not occur in bread crumb and is much cheaper than stable isotope-labeled lactic acid. The latter is relevant in the case of lactic acid because it is highly abundant in bread crumb, and therefore, a comparatively high addition of internal standard was required. The quantifier MRM transitions of all derivatized analytes and internal standards are given in Figure 4.

For validation experiments, the exhaustively extracted bread crumb residue (cf. Section 2.3) and a starch-desalted egg white powder-mixture [27] were spiked with two levels of all analytes, extracted, derivatized, and analyzed using UHPLC–MS/MS. Recovery rates were determined in a range of 84.5–119.8% (Table 5). The relative standard deviations (RSD) ranged between 1.2% and 8.1%, demonstrating the sufficient precision of the method. LODs of 0.2–174.1 nmol/kg and LOQs of 0.7–696.3 nmol/kg were determined (Table 5). The LODs and LOQs of all analytes were lower than the individual taste or aroma threshold concentrations, with the exception of the LOQs of (*E*,*E*)-2,4-decadienal (**4**, 2.1 nmol/kg) and 3-methylbutanal (**10**, 9.3 nmol/kg) which were slightly higher than the corresponding odor thresholds in water (1.3 nmol/kg; 4.6 nmol/kg) [50] but lower than the odor thresholds in starch (17.7 nmol/kg; 371.5 nmol/kg) [51].

Due to the fast extraction using bead beater homogenization, the quick derivatization (30 min), and the brief UHPLC run (10 min), the presented method enables a short overall analysis time and a high-throughput quantitation. Moreover, the extraction approach using a bead beater homogenizer needs only small sample amounts starting from 0.1 g. In addition, the extraction was performed with a methanol/water-mixture whereby considerably less starch and proteins are dissolved than with water extraction, which makes Carrez clarification redundant. A major advantage of this method is that both the nonvolatile tastants and the volatile odorants in bread crumb could be quantified simultaneously in a single UHPLC-MS/MS method. This approach of “unified flavor quantitation” was first introduced by Hofstetter et al. for key tastants and odorants in apple juice [25] and could be successfully transferred, extended, and verified for the key tastants and odorants in bread crumb. Due to its broad applicability to all kinds of carbonyl compounds, the approach could be used for the quantification of nonvolatiles and volatiles in additional foods.

## 4. Conclusions

Flavor recombination experiments elucidated sodium, potassium, ammonium, chloride, magnesium, calcium, d-fructose, l-glutamic acid, acetic acid, and lactic acid as the key taste compounds as well as acetic acid, butyric acid, vanillin, 3-methylbutyric acid, hexanal, 2,3-butanedione, phenylacetaldehyde, 3-methylbutanal, methional, (*E*,*E*)-decadienal, and (*E*)-2-nonenal as the key aroma compounds in sourdough bread crumb. Including all essential flavor compounds except for minerals, an accurate and fast UHPLC–MS/MS method using SIDA was developed and validated. For the first time, the present method enables a unified high-throughput quantitation of both the volatile key odorants and the nonvolatile key tastants of sourdough bread crumb with the same instrumental setup. This method can be used to create a kind of blueprint of the valuable aroma and taste compounds of breads in order to be independent of sensory tests in the future, especially during the industrial production of breads, and to obtain objective statements on the aroma and taste quality of breads.

## Figures and Tables

**Figure 1 foods-11-02325-f001:**
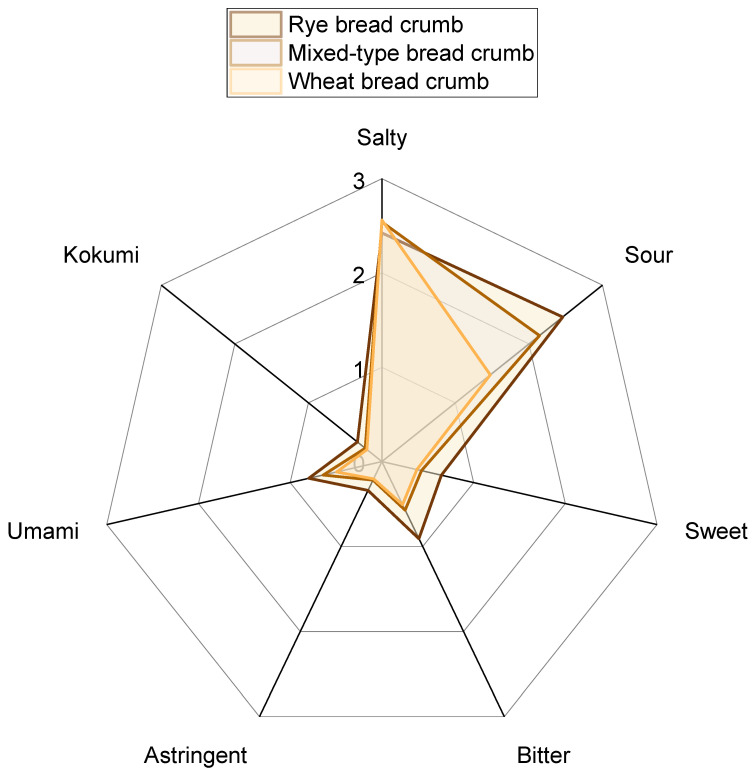
Comparative taste profile analysis of the crumbs of rye bread, mixed-type bread, and wheat bread. The intensity of each taste quality was rated on a linear scale from 0 (not perceivable) to 5 (strongly perceivable) by the trained sensory panelists (*n* = 16 for rye bread crumb, *n* = 17 for mixed-type and wheat bread crumb).

**Figure 2 foods-11-02325-f002:**
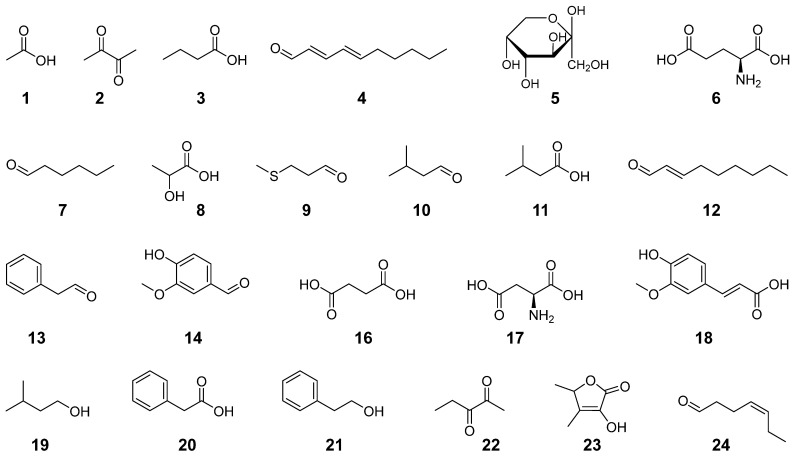
Chemical structures of the tastants and odorants present in the total taste and aroma recombinant of sourdough rye bread crumb: acetic acid (**1**), 2,3-butanedione (**2**), butyric acid (**3**), (*E*,*E*)-2,4-decadienal (**4**), d-fructose (**5**), L-glutamic acid (**6**), hexanal (**7**), lactic acid (**8**), methional (**9**), 3-methylbutanal (**10**), 3-methylbutyric acid (**11**), (*E*)-2-nonenal (**12**), phenylacetaldehyde (**13**), vanillin (**14**), succinic acid (**16**), l-aspartic acid (**17**), ferulic acid (**18**), 3-methylbutanol (**19**), phenylacetic acid (**20**), 2-phenylethanol (**21**), 2,3-pentanedione (**22**), sotolon (**23**), and (*Z*)-4-heptenal (**24**).

**Figure 3 foods-11-02325-f003:**
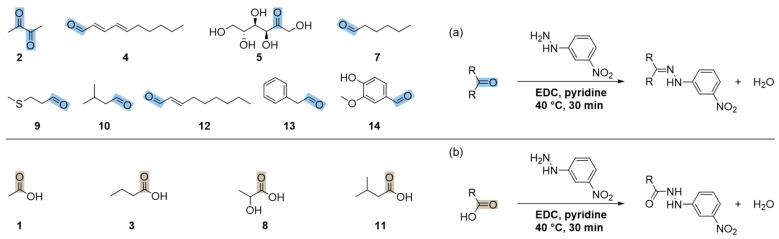
The derivatization of aldehydes, ketones (**a**), and organic acids (**b**) with 3-nitrophenylhydrazine (3-NPH) using *N*-(3-(dimethylamino)propyl)-*N*′-ethylcarbodiimide (EDC) as a coupling reagent and pyridine as a catalyst: 2,3-butanedione (**2**), (*E*,*E*)-2,4-decadienal (**4**), d-fructose (**5**), hexanal (**7**), methional (**9**), 3-methylbutanal (**10**), (*E*)-2-nonenal (**12**), phenylacetaldehyde (**13**), vanillin (**14**), acetic acid (**1**), butyric acid (**3**), lactic acid (**8**), and 3-methylbutyric acid (**11**). The carbonyl groups of aldehydes as well as ketones are colored blue and the carbonyl groups of organic acids are colored light brown.

**Figure 4 foods-11-02325-f004:**
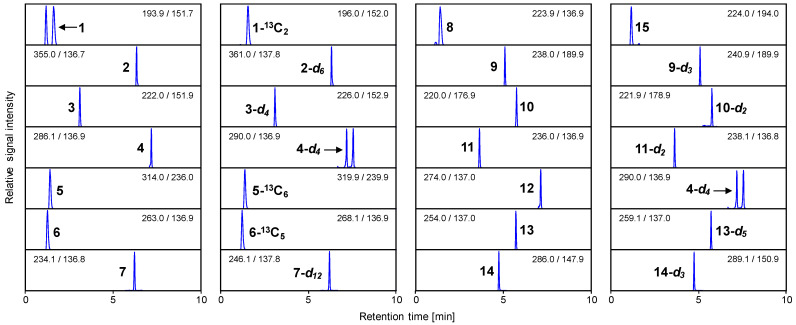
UHPLC–MS/MS mass transitions of the 3-NPH derivatized analytes and their corresponding internal standards: acetic acid (**1**), 2,3-butanedione (**2**), butyric acid (**3**), (*E*,*E*)-2,4-decadienal (**4**), d-fructose (**5**), l-glutamic acid (**6**), hexanal (**7**), lactic acid (**8**), methional (**9**), 3-methylbutanal (**10**), 3-methylbutyric acid (**11**), (*E*)-2-nonenal (**12**), phenylacetaldehyde (**13**), vanillin (**14**), acetic acid-^13^C_2_ (**1**-^13^C_2_), 2,3-butanedione-*d_6_* (**2**-*d_6_*), butyric acid-*d_4_* (**3**-*d_4_*), (*E*,*E*)-2,4-decadienal-*d_4_* (**4**-*d_4_*), d-fructose-^13^C_6_ (**5-**^13^C_6_), l-glutamic acid-^13^C_5_ (**6**-^13^C_5_), hexanal-*d_12_* (**7**-*d_12_*), 3-hydroxypropionic acid (**15**), methional-*d_3_* (**9**-*d_3_*), 3-methylbutanal-*d_2_* (**10**-*d_2_*), 3-methylbutyric acid-*d_2_* (**11**-*d_2_*), phenylacetaldehyde-*d_5_* (**13**-*d_5_*), and vanillin-*d_3_* (**14**-*d_3_*). The signal intensity of each mass transition was normalized.

**Table 1 foods-11-02325-t001:** Analyzed sourdough breads of Ludwig Stocker Hofpfisterei GmbH (Munich, Germany).

Short Name	Trade Name	Ingredients	Cereal Ratio in %	
			Rye	Wheat
Rye bread	Pfister Öko-Wilde Kruste	Flour, water, NaCl	100	0
Mixed-type bread	Pfister Öko-1331	Flour, water, NaCl	60	40
Wheat bread	Pfister Öko-Weizenlaib	Flour, water, NaCl	4	96

**Table 2 foods-11-02325-t002:** Taste qualities, taste thresholds, concentrations, and dose-over-threshold (DoT) factors of the basic tastants in the crumb of sourdough rye bread, mixed-type bread, and wheat bread.

Taste Compound	TC ^a^ [mmol/kg]	Concentration (±SD) [mmol/kg] in Bread Crumb			DoT Factor in Bread Crumb		
		Rye	Mixed-Type	Wheat	Rye	Mixed-Type	Wheat
**Group I: Salty-tasting compounds**							
Sodium	7.5 ^b^	188.7 (±2.5)	210.1 (±23.3)	228.5 (±14.3)	25.2	28.0	30.5
Potassium	13.0 ^c^	41.3 (±7.3)	32.4 (±2.4)	26.4 (±1.3)	3.2	2.5	2.0
Ammonium	5.0 ^d^	6.36 (±0.59)	4.58 (±0.36)	4.73 (±0.37)	1.3	0.9	0.9
Chloride	7.5 ^b^	193.3 (±10.7)	252.4 (±15.8)	240.2 (±4.6)	25.8	33.7	32.0
Phosphate	7.5 ^d^	16.9 (±0.6)	17.0 (±0.4)	11.9 (±0.3)	2.3	2.3	1.6
**Group II: Bitter-tasting compounds**							
Magnesium	4.0 ^d^	17.0 (±1.4)	11.5 (±0.9)	10.3 (±0.7)	4.3	2.9	2.6
Calcium	6.2 ^c^	6.60 (±1.18)	4.07 (±0.13)	4.26 (±0.31)	1.1	0.7	0.7
**Group III: Sweet-tasting compounds**							
d-Fructose	5.0 ^b^	21.4 (±0.4)	14.2 (±0.5)	9.69 (±0.20)	4.3	2.8	1.9
**Group IV: Umami-tasting compounds**							
Succinic acid	0.7 ^b^	1.39 (±0.18)	1.11 (±0.24)	1.08 (±0.24)	2.0	1.6	1.5
l-Glutamic acid	1.1 ^c^	1.28 (±0.05)	0.77 (±0.05)	0.28 (±0.02)	1.2	0.7	0.3
l-Aspartic acid	4.0 ^b^	2.44 (±0.27)	1.72 (±0.08)	0.69 (±0.05)	0.6	0.4	0.2
**Group V: Sour-tasting compounds**							
Acetic acid	1.99 ^e^	41.5 ^f^ (±2.0)	20.7 ^f^ (±0.6)	12.1 ^f^ (±1.1)	20.8	10.4	6.1
Lactic acid	14.0 ^b^	78.5 ^f^ (±1.6)	52.2 ^f^ (±2.7)	42.2 ^f^ (±3.6)	5.6	3.7	3.0
**Group VI: Astringent-tasting compounds**							
Ferulic acid	0.067 ^e^	0.069 (±0.005)	0.045 (±0.001)	0.016 (±0.001)	1.0	0.7	0.2

^a^ TC = taste threshold concentrations were determined in water, ^b^ reference [21], ^c^ reference [20], ^d^ reference [42], ^e^ reference [19], ^f^ due to losses during solvent removal using rotary evaporation and lyophilization, the concentrations of acetic acid (RBCE: 7.37 ± 0.75 mmol/kg; MBCE: 1.56 ± 0.12 mmol/kg; WBCE: 0.75 ± 0.03 mmol/kg) and lactic acid (RBCE: 66.1 ± 0.7 mmol/kg; MBCE: 43.7 ± 0.5 mmol/kg; WBCE: 37.0 ± 0.4 mmol/kg) in bread crumb extract (BCE) were lower than in the bread crumb.

**Table 3 foods-11-02325-t003:** Omission experiments performed with the taste recombinant of rye bread crumb using triangle tests.

Omitted Tastants	DoT Factor in Rye Bread Crumb Extract	Taste Quality	*p* Value	Significance ^a^
Potassium ^b^	3.2	Salty	0.004	**
Ammonium ^b^	1.3	Salty	0.019	*
Phosphate ^b^	2.3	Salty	0.382	NS
Magnesium ^b^	4.3	Bitter	0.035	*
Calcium ^b^	1.1	Bitter	0.035	*
d-Fructose ^b^	4.3	Sweet	0.017	*
Succinic acid ^b^	2.0	Umami	0.382	NS
l-Glutamic acid ^b^	1.2	Umami	0.035	*
l-Aspartic acid ^b^	0.6	Umami	0.104	NS
Succinic acid and l-aspartic acid ^b^	2.0, 0.6	Umami	0.310	NS
Lactic acid ^b^	4.7	Sour	0.009	**
Acetic acid ^b^	3.7	Sour	0.004	**
Ferulic acid ^b^	1.0	Astringent	0.448	NS
Phosphate, succinic acid, l-aspartic acid, and ferulic acid ^b–d^			0.596 ^b^; 0.596 ^c^; 0.661 ^d^	NS ^b–d^

^a^ NS = not significant (*p* > 0.05), * significant (0.05 ≥ *p* > 0.01), ** highly significant (0.01 ≥ *p* > 0.001), ^b^ the omission test was carried out in aqueous solution in comparison with the total taste recombinant (tRec_rye_), ^c^ the omission test was carried out with spiking of acetic and lactic acid in aqueous solution (pRec^+^_rye_) in comparison to the extended total taste recombinant (tRec^+^_rye_), ^d^ the omission test was carried out with spiking of acetic and lactic acid in matrix (pRec^+^_rye_(matrix)) in comparison to the extended total taste recombinant in matrix (tRec^+^_rye_(matrix)).

**Table 4 foods-11-02325-t004:** Odor qualities, odor thresholds, concentrations, and odor activity values in water (*OAV*_water_) and starch (*OAV*_starch_) of key aroma compounds in the crumb of rye bread.

Odorant	Odor Quality	Concentration [µg/kg]	TC ^a^ in Water [µg/kg]	*OAV* _water_	TC ^a^ in Starch [µg/kg]	*OAV* _starch_
Acetic acid	Pungent, sour	1,700,000	22,000 ^b^	77.3	31,140 ^b^	54.6
Butyric acid	Sweaty	2200	1000 ^b^	2.2	100 ^c^	22.0
Vanillin	Vanilla-like	1200	25 ^b^	48.0	4.6 ^c^	261
3-Methylbutyric acid	Sweaty	1100	740 ^c^	1.5	5.5 ^c^	200
Hexanal	Green, grassy	380	10.5 ^b^	36.2	30 ^b^	12.7
2,3-Butanedione	Buttery	356	15 ^b^	23.7	6.5 ^c^	54.7
Phenylacetaldehyde	Honey-like	244	4.0 ^b^	61.1	28 ^d^	8.7
3-Methylbutanal	Malty	151	0.4 ^b^	378	32 ^c^	4.7
Methional	Cooked potato-like	75	1.8 ^b^	41.7	0.27 ^c^	278
(*E*,*E*)-2,4-Decadienal	Fatty, waxy	62	0.2 ^b^	310	2.7 ^c^	23.0
(*E*)-2-Nonenal	Green, fatty	49	0.8 ^b^	61.3	0.53 ^c^	92.5

^a^ TC = Odor threshold concentrations, ^b^ reference [50], ^c^ reference [51], ^d^ reference [2].

**Table 5 foods-11-02325-t005:** Validation experiments for the quantitation of 3-NPH-derivatized key taste and aroma compounds in bread crumb.

No. ^a^	Low-Spiking Experiment				High-Spiking Experiment				LOD ^e^ [µmol/kg]	LOQ ^f^ [µmol/kg]	TC ^g^ (Taste) [µmol/kg]	TC ^g^ (Odor) [µmol/kg]
	Bread Crumb Matrix		Starch-Egg White-Matrix		Bread Crumb Matrix		Starch-Egg White-matrix					
	RSD ^b,d^ in %	Recov. ^c,d^ in %	RSD ^b,d^ in %	Recov. ^c,d^ in %	RSD ^b,d^ in %	Recov. ^c,d^ in %	RSD ^b,d^ in %	Recov. ^c,d^ in %				
1	1.9	104.0	1.9	107.4	4.1	101.5	1.6	106.2	0.174	0.696	1990 ^h^	366 ^i^
2	5.2	108.2	1.4	117.8	8.1	114.7	2.5	111.6	0.011	0.032	-	0.174 ^i^
3	2.8	117.2	1.7	98.9	3.6	118.8	2.1	114.4	0.008	0.031	4000 ^j^	11.3 ^i^
4	3.5	107.3	4.3	97.6	1.9	114.6	5.4	110.9	0.0005	0.002	-	0.001 ^i^
5	1.8	102.2	2.2	102.4	4.7	99.5	3.6	105.0	0.023	0.093	5000 ^k^	-
6	1.4	106.1	2.7	106.9	4.9	103.2	2.1	108.0	0.001	0.003	1100 ^j^	-
7	4.0	99.2	5.0	107.7	4.0	98.9	1.7	113.1	0.0006	0.002	-	0.105 ^i^
8	6.7	112.6	3.8	113.3	6.5	103.7	5.6	112.2	0.052	0.189	14,000 ^k^	-
9	1.6	118.5	2.0	114.1	3.2	117.6	1.4	117.4	0.002	0.007	-	0.017 ^i^
10	3.3	107.9	2.3	85.3	6.7	105.9	5.3	109.2	0.002	0.009	-	0.005 ^i^
11	2.8	108.0	2.2	87.1	3.8	116.5	2.4	99.7	0.001	0.003	-	7.25 ^l^
12	2.9	103.4	3.1	88.0	3.2	116.4	4.0	84.5	0.001	0.004	-	0.006 ^i^
13	4.2	116.8	1.9	118.2	4.1	119.8	1.2	117.0	0.0002	0.0007	-	0.033 ^i^
14	1.4	114.7	2.6	111.5	2.9	116.5	1.2	111.5	0.0003	0.001	-	0.164 ^i^

^a^ Names of the numbered analytes: acetic acid (**1**), 2,3-butanedione (**2**), butyric acid (**3**), (*E*,*E*)-2,4-decadienal (**4**), d-fructose (**5**), l-glutamic acid (**6**), hexanal (**7**), lactic acid (**8**), methional (**9**), 3-methylbutanal (**10**), 3-methylbutyric acid (**11**), (*E*)-2-nonenal (**12**), phenylacetaldehyde (**13**), and vanillin (**14**), ^b^ RSD = relative standard deviation, ^c^ Recov. = recovery, ^d^ data are the means of replicated sample work-ups (*n* = 3) and analyses (*n* = 2), ^e^ LOD = limit of detection, ^f^ LOQ = limit of quantitation, ^g^ TC = threshold concentration determined in water, ^h^ reference [19], ^i^ reference [50], ^j^ reference [20], ^k^ reference [21], ^l^ reference [51].

## Data Availability

Data are contained within the article and Appendix A.

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
