# Peer review of "The Sensory-Directed Elucidation of the Key Tastants and Odorants in Sourdough Bread Crumb"

_foods, 2022, doi:10.3390/foods11152325_

Round 1

Reviewer 1 Report

The authors present a two part study, where they first identify taste and odor components in breads and then develop a UHPLC-MS/MS methodology to quantify all key components simultaneously.

Comments:

1) Breads should be described in more detail. E.g. what bacteria were used for sourdough preparation? How exactly were the doughs prepared? (times, T, etc.). Were the same amounts of salt used for each bread?

2) Please explain why the preparation for BCE is different to sample preparation. Mainly, why a different sample to volume ratio was used for extraction.

3) Table 2: Please add "Rye", "Mixed-type" and "Wheat" under "Concentration in bread crumb" for improved understanding.

4) Regarding 3.3: The first paragraph does not belong in results & discussion. Please move to another section

5) I suggest that figure 3 is moved to supplementary.

6) Compounds were first quantified using various methods and then a UHPLC-MS/MS approach using SIDA was developed to quantify aroma and taste compounds simultanesouly. However, a comparison of the results of these varying quantification methods is missing. Please add them.

7) Conclusion should be revised to give a more detailed conclusion of the conducted study.

Author Response

Review No. 1:

The authors present a two part study, where they first identify taste and odor components in breads and then develop a UHPLC-MS/MS methodology to quantify all key components simultaneously.

Comments:

  • Breads should be described in more detail. E.g. what bacteria were used for sourdough preparation? How exactly were the doughs prepared? (times, T, etc.). Were the same amounts of salt used for each bread?

Response: Thank you for asking this interesting questions. As detailed in section 2.2, all breads were commercially purchased from the Ludwig Stocker Hofpfisterei GmbH (Munich, Germany). The company produces the breads in a traditional three-stage sourdough bread-making process. This means that no lactic acid bacteria or yeast is added. Unfortunately, the specific production parameters are a company secret (fermentation/baking time and temperature, …). The company indicates only average sodium chloride contents of the final breads with 1.5 g/100 g for the mixed-type bread and wheat bread and 1.2 g/100 g for the rye bread. The analytically determined concentrations of sodium and chloride in the bread crumbs are listed in Table 2.

Changes: In line 107-108: We pointed out that the three-stage sourdough bread-making is a traditional process without addition of lactic acid bacteria or yeast. In line 109-110: Clarification that not only the rye bread but also the mixed-type and wheat bread were produced using this three-stage sourdough bread-making process.

  • Please explain why the preparation for BCE is different to sample preparation. Mainly, why a different sample to volume ratio was used for extraction.

Response: Thank you for your comment. The preparation of the bread crumb extract (BCE) is a large-scale extraction. Using this preparation, a sufficient amount of BCE is gained for the sensory analysis. The sample preparation using a Precellys bead beater homogenizer is a small-scale extraction. It enables a stable isotope dilution analysis because just small amounts of internal standards have to be added (saves resources and money). This kind of extraction would not be sufficient to yield enough extract for sensory analysis. Because of this, both extraction procedures are needed. The sample to volume ratio of the BCE work-up is higher compared with that of the Precellys work-up. In pretests, it was shown that both work-ups are sufficient to extract the main portion of all taste-active compounds. The remaining percentage of tastants that was extracted with the 3x H2O-extraction after the 3x MeOH/H2O-extraction in the BCE work-up was not taste-active. For this reason and to save resources, less solvent was used in case of the BCE work-up.

  • Table 2: Please add "Rye", "Mixed-type" and "Wheat" under "Concentration in bread crumb" for improved understanding.

Response: Thank you for pointing this out. We have changed it as recommended.

Changes: A new row with "Rye", "Mixed-type" and "Wheat" was added in Table 2 under "Concentration in bread crumb".

  • Regarding 3.3: The first paragraph does not belong in results & discussion. Please move to another section

Response: Thank you for your comment. We agree that the main part of the paragraph does not belong there. We moved it as recommended.

Changes: The second, third and fourth sentence of the paragraph was moved to the introduction. The last two sentences of the paragraph discuss the advantages of this method. Therefore, it was moved to the end of section 3.3 because there the other advantages are explained. Because of these adjustments, the numbers of the references have changed in the manuscript and also in the supplementary information.

  • I suggest that figure 3 is moved to supplementary.

Response: Thank you for your suggestion. We think that figure 3 is needed in the main manuscript to understand the general principle of the 3-NPH derivatization and the specific reaction with the key taste and aroma compounds for the novel “unified flavor quantitation method”.

  • Compounds were first quantified using various methods and then a UHPLC-MS/MS approach using SIDA was developed to quantify aroma and taste compounds simultanesouly. However, a comparison of the results of these varying quantification methods is missing. Please add them.

Response: Thank you for your comment. We agree that this is an interesting point. Due to the length of the current manuscript, we focused on the development and validation of the method. In our further research, we intend to carry out such a comparison and an application of the presented method on a wide range of breads. 

  • Conclusion should be revised to give a more detailed conclusion of the conducted study.

Response: Thank you for pointing this out.

Changes: We revised the conclusion.

Reviewer 2 Report

This experiment illustrated the key aroma compounds in sourdough bread crumb using flavor recombination experiments. An accurate and fast UHPLC-MS/MS method using SIDA was also developed and validated. I have a question that would like to be clarified.

In the comparative taste profile analysis, why only 16 trained sensory panelists rated rye bread crumbs, while the other two were evaluated by 17? 

Author Response

Review No. 2:

This experiment illustrated the key aroma compounds in sourdough bread crumb using flavor recombination experiments. An accurate and fast UHPLC-MS/MS method using SIDA was also developed and validated. I have a question that would like to be clarified.

In the comparative taste profile analysis, why only 16 trained sensory panelists rated rye bread crumbs, while the other two were evaluated by 17?

Response: Thank you for your comment. Due to the Covid-19 pandemic just 16 panelists were available for this specific sensory analysis.

Reviewer 3 Report

The manuscript submitted by Frank and Hofmann et al. mainly dealt with sensory-directed elucidation of the key tastants and odorants in sourdough bread crumb. The paper is very interesting and instructive. Some errors or problems need to be explained or revised according to the following suggestions.

Abstract: 

none should be revised.

Introduction

introduction should be focused on soughdough product or bakery product. 

line 50 to 54, I think authors don't need to list so many kinds of food. 

Introduction should find the focus of this MS and also strengthen of the significance of this study.

Materials and methods

none should be revised.

Results and discussion

none should be revised.

Conclusion

limitations of this study and further research plan should be added.

Author Response

Review No. 3:

The manuscript submitted by Frank and Hofmann et al. mainly dealt with sensory-directed elucidation of the key tastants and odorants in sourdough bread crumb. The paper is very interesting and instructive. Some errors or problems need to be explained or revised according to the following suggestions.

  • Abstract:

None should be revised.

  • Introduction:
  • Introduction should be focused on soughdough product or bakery product.

Response:

Since the main focus of the manuscript is on sensory- and quantitative analysis of aroma and flavor active compounds and not on the technology of baking sourdough breads or other baked goods (only commercially available breads were analyzed, and we had no influence on their composition or technological parameters), we would like to leave the introduction as it is.

  • Line 50 to 54, I think authors don't need to list so many kinds of food.

Response:

Changed as recommended, the list has been significantly shortened.

 Introduction should find the focus of this MS and also strengthen of the significance of this study.

Response: 1) We added a paragraph regarding the advantages of the unified flavor quantitation because the application of this method is one of the main goals of the manuscript. 2) We add a sentence at the end of the introduction to make the significance of the study more clear.

  • Materials and methods:

None should be revised

  • Results and discussion:

None should be revised.

  • Conclusion:

Limitations of this study and further research plan should be added.

Response: We add a sentence to ….the conclusion, from our point of view there are no limitations, quite the contrary, the method could be adopted to all kind of foods. There is no further research planned.

Changes: This method can be used to create a kind of blueprint of the valuable aroma and flavor compounds of breads in order to be independent of sensory tests in the future, especially during the industrial production of breads, and to obtain objective statements on the aroma and flavor quality of breads.